# Occupational Noise-Induced Hearing Loss among Migrant Workers in Kuwait

**DOI:** 10.3390/ijerph18105295

**Published:** 2021-05-16

**Authors:** Mariam Buqammaz, Janvier Gasana, Barrak Alahmad, Mohammed Shebl, Dalia Albloushi

**Affiliations:** 1Department of Environmental & Occupational Health, Faculty of Public Health, Kuwait University, Hawalli 13110, Kuwait; janvier.gasana@gmail.com (J.G.); B.Alahmad@g.harvard.edu (B.A.); 2Department of Occupational Health, Ministry of Health, Shuwaiba Industrial Medical Center, Ahmadi 60000, Kuwait; mshebl2000@hotmail.com; 3Mubarak Al-Kabeer Hospital (MKH), Ministry of Health, Hawalli 47060, Kuwait; daliaalbloushi@gmail.com

**Keywords:** migrants, noise, occupational noise induced hearing loss

## Abstract

Although the effect of hearing loss on years lived with disability (YLD) is quite substantial, occupational hearing loss among migrant workers is significantly under-studied. In Kuwait, where nearly two-thirds of the population are migrant workers, the burden of occupational noise-induced hearing loss (ONIHL) is unknown. The objective of the study was to assess the prevalence of ONIHL among migrant workers in Kuwait and explore workplace and individual risk factors that are associated with ONIHL. We obtained data of annual physical exams for the year 2018 conducted by the Shuaiba Industrial Medical Center (SIMC) for all industrial workers in the area. We applied univariate and multivariate logistic regression models to estimate the effects of individual and occupational characteristics on ONIHL. A total of 3474 industrial workers visited the SIMC for an annual exam. The vast majority were men (99%) and non-Kuwaitis (98%) with a median age of 38 years. A total of 710 workers were diagnosed with ONIHL with a prevalence of 20.4%. Age, years of experience, and self-reported exposure to noise were associated with statistically significant higher odds of ONIHL. When adjusted for age, years of experience, and other individual level factors, type of industry was not a statistically significant predictor of ONIHL. The study uncovers the significant burden of hearing loss among the migrant worker subpopulation in Kuwait, an area of occupational health that is often underestimated or unrecognized. Although laws and regulations are in place to prevent and control noise in the workplace, the onus is on local authorities to ensure the necessary training and controls aimed to reduce noise exposure.

## 1. Introduction

Noise in the environment ranges from tolerable to hazardous levels. According to the International Labour Organization (ILO), noise is linked to workplace environmental pollution which may lead to adverse health effects and burden the economy [1]. Continuous noise exposure raises levels of stress and lowers quality of living [2,3]. The safe noise threshold is 85 decibels (dB), over which a person’s hearing may be damaged with prolonged exposure [4]. At the international scale, hearing loss is increasing along with the increase in aging populations. Interestingly, occupational noise-induced hearing loss (ONIHL) is one of the most prevalent work-related diseases [5], such that approximately 16% of adult-disabling hearing loss is attributed to occupational noise [6]. In the US, an estimated 12% of the American workforce population suffers from hearing loss, out of which 24% is due to work-related exposures [7].

Hearing loss is a significant public health problem that denotes social and economic burdens [8]. Although the notion that people do not die directly from hearing loss is true, the effect of hearing loss on years lived with disability (YLD) is quite substantial. Hearing loss is one of the top five causes of global YLD and is associated with depression, cognitive decline, dementia, risk of falls, and hospitalizations [9,10]. Additionally, companies pay approximately $242 million yearly as compensation for the hearing loss disabilities of workers in the US [11].

The prevalence of occupational hearing loss in developing countries at 23% is more problematic compared with developed countries at 16% [12]. For example, the prevalence and burden of workers in Kuwait with hearing loss due to occupational exposure are unknown. Consequently, prevention and control measures are not being sufficiently considered and implemented. Meanwhile, the majority of Kuwait’s workforce are male migrant workers employed in low-skilled sectors [13]. In general, migrant workers experience high levels of hazardous job-related exposure in their working environments, which results in negative effects on health, such as work-related injuries and disabilities [14]. According to [15], hearing loss in migrant workers is significantly under-studied. Workplace hazards can be especially amplified among migrant workers due to language and cultural barriers. Typically, migrant workers are marginalized subpopulations frequently employed in the least desired professions with a high risk of occupational injury [16].

ONIHL among migrant workers is an overlooked public health problem in Kuwait, which necessitates urgent policy and regulatory implementation. To bridge the research gap, the current study aims to investigate the prevalence of ONIHL among migrant workers in Kuwait and explore workplace and individual risk factors that are associated with ONIHL.

## 2. Subjects and Methods

### 2.1. Study Population

This cross-sectional study is retrospective in nature and utilizes data from annual medical examinations conducted by the Shuaiba Industrial Medical Center (SIMC) at the Department of Occupational Health of the Ministry of Health. The SIMC is located in the Al Shuaiba industrial area and provides services to migrant workers from various industrial factories and corporations located in the Al Ahmadi governorate [17]. The SIMC collects data from periodical, pre-employment, and medical fitness tests for all industrial workers in the area. The study obtained data for all physical exams for the year 2018 (January to December). The sample consists of workers from various nationalities and aged over 21 years. Data were considered secondary because information was not collected specifically for the study. Ethical approval was obtained from the Ministry of Health and Kuwait University with only the primary researchers granted access to anonymized data.

### 2.2. Audiometry Tests

All audiometric tests were conducted by two experienced, well-trained nurses in a testing facility fulfilling ISO 8253-1(1989) criteria. All subjects were examined by otoscopy to exclude external and middle ear medical disorders such as ear wax, otitis externa, and otitis media, followed by Pure-tone audiometric tests (air and bone conduction) that were conducted to determine the hearing thresholds across the conventional frequencies 250, 500, 1000, 2000, 3000, 4000, 6000, and 8000 Hz for both ears of each subject, using a Madsen-Orbiter 922: Diagnostic Audiometer, with TDH-50P earphones. The audiometer met ANSIS3.26-1981 standards and was calibrated. Measurements were taken using 5 dB increments. Audiometric tests were only made at least 18 h after the last exposure to noise to allow recovery from any temporary hearing threshold shifts. Noise-induced hearing loss is a “sensorineural hearing deficit that begins at the higher frequencies (3000 to 6000 Hz) and develops gradually as a result of chronic exposure to excessive sound levels” [18]. Occupational noise-induced hearing loss is defined as hearing loss due to unremitting or intermittent noise exposure in the work environment, it is bilateral, and recognized by a notch shape in the audiogram at 3000, 4000, or 6000 Hz [19]. We calculated noise-induced hearing loss as the average of a hearing threshold level for the critical noise-sensitive frequencies (3000, 4000, and 6000 Hz) with a 25 dB threshold [20,21]. Only when NIHL was bilateral was it classified as occupational NIHL. Typically, ONIHL is bilateral due to symmetrical exposure to noise, whereas unilateral NIHL can be attributed to many reasons aside from occupation [19]. Workers with missing audiometric results and exposure data were excluded from analysis.

### 2.3. Individual Characteristics

Most of the workers on average worked 72 h per week (12 h per day, 6 days a week). However, official daily working hours per day were 8, and the extra hours were considered as overtime earning additional income. Moreover, SIMC conducted a survey on self-reported noise exposure (yes/no), which was completed by the participants. The questionnaire also collected data on age, gender, years of experience, noise exposure, nationality, job type, and industry type.

### 2.4. Statistical Analysis

Continuous variables were expressed as mean and standard deviations or median and interquartile ranges. Categorical variables, such as age group, gender, years of experience (≤15, 16–29, and ≥30 years), noise exposure (yes/no), nationality, job type, and industry type were presented as numbers and percentages. To examine the independent factors associated with ONIHL, the study employed univariate and multivariate logistic regressions and reported odds ratios with confidence intervals for each factor. Univariate analyses were conducted to investigate the relationship between the ONIHL and other predictors, one at a time. Multivariate logistic regression was performed to elucidate the relationship between the outcome and variables after adjusting for other predictors. A significance level of *p* < 0.05 was used for all tests. Statistical procedures were performed using SPSS version 26 (SPSS, IBM Corp. Armonk, NY, USA).

## 3. Results

Table 1 presents the characteristics of 3474 industrial workers who visited the SIMC in 2018 and the prevalence of ONIHL. The vast majority were male (98.8%) with a median age of 38 years (IQR; 15 years). A total of 710 workers were diagnosed with ONIHL with prevalence rates of 20.4% and 20.6% for the entire population and amongst males, respectively. According to age, the highest prevalence rate was observed among the ≥61 age group (69.6%). Kuwaitis represented 2% of the sample, with a prevalence rate of 17.1%. When stratified by nationality, high prevalence rates were noted among non-Kuwaiti workers, especially those from Pakistan (36.9%) and the Philippines (28.2%). Based on years of tenure, those who had worked for 30 years or more displayed a high rate of ONIHL (63.3%). In terms of self-reported noise exposure, the prevalence rate of ONIHL among those who answered “yes” was 29.0% compared with 15.8% among those who answered “no”.

Table 2 presents ONIHL by job and industry type. The highest prevalence rates were observed for services and sales workers (31.8%) followed by crafts and related trade workers (29.0%). According to industry type, workers in transportation and storage displayed the highest prevalence (35.7%). Stratification by noise exposure is reported in the Appendix A.

### Regression Analyses

Table 3 reports the unadjusted and adjusted odds ratios of ONIHL. Crude odds ratios suggest that age, years of experience, and self-reported exposure to noise are associated with higher odds of ONIHL. Compared with young workers (21–30 years), workers aged 31–40 year and ≥61 years exhibited nearly twice the odds and an approximately 40-fold increase in the crude odds of ONIHL. Work experience of ≥30 years is associated with unadjusted odds ratios of 7.9 (95% CI: 4.4–14.4) compared to those with ≤15 years. Compared to Indians (i.e., the nationality with the greatest number of workers), univariate analysis revealed that other nationalities were significantly associated with higher ONIHL, such as Filipinos (OR 1.5; 95% CI: 1.1–2.2), Pakistanis (OR: 2.4; 95% CI: 1.7–3.4), and Bangladeshis (OR: 1.2; 95% CI: 0.87–1.6). When investigating the relationship between job type and ONIHL, the study found that managers (OR: 2.3; 95% CI: 1.1–4.8), services and sales (OR: 2.6; 95% CI: 1.1–6.6), and plant and machine operators and assemblers (OR: 1.8; 95% CI: 1.4–2.3) displayed statistically significant higher odds of ONIHL compared with elementary occupations. Finally, except for the construction and wholesale and retail trade and repair of vehicles and motorcycles industries, significantly lower odds of ONIHL were noted for all other industries compared with administrative and supportive service activities.

After adjustment, the odds of ONIHL among age groups remained statistically significantly higher than the reference group, although the effect estimates were attenuated after adjustment. Workers aged >61 years were 30.5 times more likely to have ONIHL than young workers. A trend is clearly observed: the odds of ONIHL increase for every 10 years increase in the categories of age. The coefficient for gender is no longer statistically significant in the multivariate regression after adjusting for the other variables. A work experience of >30 years exhibits twice the adjusted odds of ONIHL compared to a work experience of ≤15 years. Similarly, self-reported exposure to noise is associated with an adjusted odds ratio of 2.0 (1.7–2.4). Furthermore, the results for Filipinos (OR: 1.5: 95% CI: 1.0–2.2) and Pakistanis (OR: 2.2; 95% CI: 1.4–3.3) remain significant compared with Indians (reference group). Finally, job and industry type became non-significant, except for job type crafts and related trade works (OR: 1.6; 95% CI: 1.2–2.2).

## 4. Discussion

To the best of our knowledge, this study is the first in Kuwait and the Gulf Region to assess the prevalence and predictors of ONIHL among migrant workers. We leveraged routinely collected secondary data from the SIMC to enable the investigation of ONIHL. The study uncovers the significant burden of hearing loss among the migrant worker subpopulations. This area of occupational health is often underestimated or unrecognized among marginalized subpopulations, which renders it an important public health topic [15].

Unwarranted exposure to noise can result in cochlear trauma which leads to hearing loss and tinnitus. Occupational workers are at most risk of noise-induced hearing loss. Noise-induced hearing loss is a term used to describe the effects of long term and continuous exposure to noise. As an exposure, magnitude of noise can be correlated to cochlear damage. Acoustic trauma is another term used to describe sudden hearing loss typically caused by single or repeated noise exposure. NIHL can be characterized by being either temporary, where the individual can regain their hearing within 48 h, known as temporary threshold shift (TTS), or it can result in permanent threshold shift (PTS). Some individuals might experience hidden hearing loss, where a pure-tone threshold shift is absent. The pathophysiology of permanent threshold shift is as follows: loss of the outer hair cells with evidence suggesting that this might cause the degeneration of the auditory nerve (as demonstrated by histopathology of the temporal bone) [22]. In a similar way, it can be said that excessive exposure to noise ultimately leads to the damage of the organ of corti. This can be compartmentalized into two main categories: the first being mechanical destruction typically caused by short exposure to noise or metabolic decompensation, a sequel of prolonged exposure to noise by reactive oxygen species, formation of free radicals and glutamate excitotoxicity [23,24]. All which ultimately lead to cell death. Equally important, noise exposure also surges the levels of free calcium existing in the outer hair cells, again, activating apoptosis and necrosis [24,25,26]. Audiometric tests are imperative for determining the degree of hearing impairment caused by noise. Firstly, pure tone audiometry recognizes the hearing threshold of an individual and is mostly used to establish the degree of NIHL. A major downfall of this investigation is the difficulty in separating presbycusis (age related hearing loss) and NIHL [24]. Secondly, a decrease in speech recognition scores alongside a normal audiogram can be associated with NIHL. Subsequently, an otoacoustic emission test is a sensitive and easy means by which to diagnose NIHL, where it can be used to detect those in the pre-symptomatic phase (early indicator) and normal audiograms. Though, it is not a useful tool when hearing loss is already present [24,27]. The last tool that can be used for determining the extent of NIHL is an objective measure for noise-induced-synaptopathy, which is an electrophysiological measurement lacking both sensitivity and evidence in humans. Additionally, tinnitus was reported to affect approximately 24% of those that have been exposed to noise in comparison to the general population where only 14% reported symptoms [28]. Finally, concurrent exposure to ototoxic substances, such as solvents and heavy metals, may also contribute to the damaging effect of noise [29,30]. The extent to which these ototoxic substances interact with noise needs to be further explored [31].

The evidence at hand shows that migrant workers in Kuwait suffer from ONIHL with a prevalence of 20.4%, which is in line with the results of [20,32] at 21.5% in Ghana and 22.1% in Germany, respectively. However, the estimated prevalence in the present study is lower than those of [33,34], at 58.5% among textile workers in Tanzania and 44% among woodworkers in Nepal, respectively. Prevalence ratios can vary depending on the study sample and the nature of the workplace; therefore, they are not readily generalizable nor easily comparable. However, as expected, the current study found that ONIHL is significantly associated with age, nationality, years of experience, noise exposure, and job type, in agreement with the previous literature [20,35]. The odds of ONIHL increased with age and years of work experience, which is in line with [35,36]. Furthermore, [37,38] found that the prevalence of ONIHL was the highest among workers in construction with more years of work experience. Because our sample is largely from the manufacturing industry, construction workers were, on average, younger and not fully representative of the entire construction industry which may explain the apparent crude protective effect of working in construction compared to administrative workers who tend to be older. After adjustments to age and years of experience, none of the industry types were significant.

The age pattern observed in the study is similar to those of [39,40]. Interventions aimed at reducing exposure to noise in the workplace must consider predictors that are significantly associated with ONIHL. For example, young workers should be targeted for risk reduction, to prevent hearing damage in older age. Such interventions include re-designing the workplace, noise training, substituting forms of work, providing community and workplace support, and implementing health promotion and disease prevention programs, as well as support programs for employees [41].

Migrant workers worldwide suffer from significant issues that compromise their physical health and mental wellbeing. In addition, they are at risk of having occupational diseases and injuries due to many stressors that arise from individual, occupational, environmental, and community domains [42]. Migrant workers experience higher rates of hazardous job-related exposures in their work environments, which significantly result in adverse health effects, work injuries, and work-related fatalities [14]. In particular, migrants in Kuwait could be synergistically exposed to occupational injuries and diseases, due to cultural and language barriers that prevent adequate safety training. Migrant workers comprise more than two-thirds of the population in Kuwait but are mostly employed in low-wage and hazardous jobs that may not provide adequate personal protective equipment and offer little or no training. Cumulatively, migrant workers become susceptible to ONIHL despite their young age.

The Kuwait Environmental Public Authority (KEPA) regulates noise control in industrial settings, namely, Articles (19), (54), and (55) from the Environmental Protection Law 42/2014 Amended by 99/2015 and Decision No. (210/2001). Furthermore, the Ministry of Social Affairs and Manpower have a Ministerial resolution No. (208/2011) regarding the safety levels and standards of noise in workplaces and areas. The legal text of each regulation is mentioned in Table 4 [43,44,45].

Although laws and regulations are in place to prevent and control noise in Kuwait, the results indicate a possible problem in the implementation of such rules and regulations. An overall prevalence of 20.4% in a slightly younger workforce indicates that industrial institutions and workers in Kuwait have a lot more to do. Thus, it is recommended that each company should provide a periodic report of its noise levels to KEPA, which should establish a robust monitoring system for industrial companies and a disciplinary fine system for violators. Moreover, each industrial company should provide employees with hearing conservation programs, which are intended to serve workers exposed to occupational noise. In general, a hearing conservation program should include noise exposure monitoring, hearing protection, and hearing testing, training, and record-keeping [46]. International companies, especially globally recognized ones, widely utilize hearing conservation programs. In several countries, these programs are mandatory. In Kuwait, however, hearing conservation programs are not legally mandated. Notably, the SIMC recently initiated its hearing conservation program and is expected to yield results in the near future. Moreover, we recommend that companies should follow optimal methods for preventing workplace hazards through the use of primary prevention measures, such as engineering controls, modification of work practices, and administrative controls. Primary prevention is the best strategy for avoiding the effects of acoustic trauma. Hearing conservation programs (HCPs) in grade school children are potentially effective to increase the knowledge about the hazards of noise exposure early in life and this may be associated with behavioral changes towards noise reduction and ear protection [47]. Such programs, however, are less applicable to migrant workers who are unlikely to have received such training in their respective countries. For noise in the workplace, the hierarchy of controls has its place here, from elimination (best line of defense) or reduction of noise through engineering or administrative controls. Laws and legislation on occupational noise exposure (if and when enforced) will be instrumental in regulating noise exposure, which will result in noise reduction and/or noise reducing technical improvements to protect employees [48]. Hearing protection equipment offers a secondary level of protection. Indeed, there are challenges in preventing and controlling exposure to occupational noise that need to be recognized in establishing a hearing conservation program. Hearing conservation programs that have policies on top of minimum compliance seem to do better in the prevention of occupational noise-induced hearing loss [49]. Ref. [50] concluded that, while earmuffs and earplugs can reduce noise to safe levels, without proper training, they might not provide sufficient protection. On the other hand, engineering solutions might be as effective as utilizing PPE, only if they are implemented carefully [50].

## 5. Limitations

The study has its limitations. First, the sample was extracted from data provided by the SIMC; thus, the sample may not be representative of all industrial workers in Kuwait. However, the results are comparable to studies utilizing random samples. Second, noise exposure information was self-reported. Such exposure assessment is prone to recall bias and therefore should be interpreted with caution. Similarly, it was not possible to separate noise exposure from work and from outside work. The current study used one audiogram per employee in estimating a snapshot prevalence. Follow-up studies based on a baseline audiogram and a repeated measure of periodic audiogram per worker will be critical to establish temporality. The stratification of subgroups may have resulted in smaller sample sizes and therefore the readers must be cautioned when interpreting the results. We therefore did not examine the severity of any hearing loss in this cohort. Moreover, the study was short of data on education level and behavioral aspects, such as the use of headphones for phone calls or listening to music, which may have influenced the results. Future research should focus on a representative sample and collect additional specific data on hearing thresholds and covariates to render the results generalizable. Finally, future studies in Kuwait must also investigate the economic and social burdens of ONIHL, which could be significant to public health.

## 6. Conclusions

In conclusion, the study provided evidence that one-fifth of migrant industrial workers in Kuwait suffer from hearing loss likely from occupational exposure. The prevalence of ONIHL calls for prevention using available measures, such as the existing use of hearing protection devices and hearing conservation programs. The results point to several recommendations. In Kuwait, companies must adhere to the permissible noise levels following the local laws as well as international standards such as the recommendations of the U.S. Occupational Safety and Health Administration (OSHA). In addition, the implementation of laws must be more assertive as KEPA continues its monitoring of the industrial sector. Preventing hearing loss can reduce significant social and health burdens on individuals and the population.

## Figures and Tables

**Table 1 ijerph-18-05295-t001:** Baseline characteristics of industrial workers according to occupational noise-induced hearing loss status.

Demographic Characteristics	N	All (*n* = 3474)		Without ONIHL (*n* = 2764)		With ONIHL (*n* = 710)		
		*n*	(%)	*n*	(%)	*n*	(%)	*p*-Value
**All**	3474	3474	(100)	2764	(79.6)	710	(20.4)	
Age (years)	3474							<0.001 *
21–30		712	(20.5)	672	(94.4)	40	(5.6)	
31–40		1268	(36.5)	1124	(88.6)	144	(11.4)	
41–50		921	(26.5)	689	(74.8)	232	(25.2)	
51–60		517	(14.9)	262	(50.7)	255	(49.3)	
≥61		56	(1.60)	17	(30.4)	39	(69.6)	
Median Age (years) (IQR)	3474	38	(15)	37	(13)	49	(14)	<0.001 *
**Gender**	3474							<0.002 *
Male		3434	(98.8)	2725	(79.4)	709	(20.6)	
Female		40	(1.2)	39	(97.5)	1	(2.5)	
**Nationality**	3474							<0.001 *
Indian		1974	(56.8)	1584	(80.2)	390	(19.8)	
Egyptian		483	(13.9)	392	(81.2)	91	(18.8)	
Bangladeshi		271	(7.8)	210	(77.5)	61	(22.5)	
Filipino		181	(5.2)	130	(71.8)	51	(28.2)	
Pakistani		141	(4.1)	89	(63.1)	52	(36.9)	
Kuwaiti		70	(2.0)	58	(82.9)	12	(17.1)	
Others		354	(10.2)	301	(85.0)	53	(15.0)	
**Experience (years)**	3441							<0.001 *
≤15		2974	(85.4)	2448	(82.3)	526	(17.7)	
16–30		418	(12.1)	272	(65.1)	146	(34.9)	
≥30		49	(1.40)	18	(36.7)	31	(63.3)	
Median Experience (years) (IQR)	3441	3	(3)	2	(4)	4	(9)	<0.001 *
**Exposure to Noise**	3474							<0.001 *
No		2262	(65.1)	1904	(84.2)	358	(15.8)	
Yes		1212	(34.9)	860	(71.0)	352	(29.0)	

ONIHL: occupational noise-induced hearing loss; * Significant at the 5% level. The categorical association is conducted using the global chi-square test. The Kruskal–Wallis test was conducted to determine the association between continuous variables.

**Table 2 ijerph-18-05295-t002:** Baseline job and industry type of industrial workers according to occupational noise-induced hearing loss status.

Job and Industry Type	N	All (*n* = 3474)		Without ONIHL (*n* = 2764)		With OHNIL (*n* = 710)		
		*n*	(%)	*n*	(%)	*n*	(%)	*p*-Value
**All**	3474	3474	(100)	2764	(79.6)	710	(20.4)	
**Job Type**	3474							<0.001 *
Managers		38	(1.1)	27	(71.1)	11	(28.9)	
Professionals		204	(5.9)	179	(87.7)	25	(12.3)	
Technicians and associate professionals		886	(25.5)	733	(82.7)	153	(17.3)	
Clerical support		110	(3.2)	98	(89.1)	12	(10.9)	
Services and sales		22	(0.6)	15	(68.2)	7	(31.8)	
Elementary occupations		735	(21.2)	624	(84.9)	111	(15.1)	
Crafts and related trade works		669	(19.3)	475	(71.0)	194	(29.0)	
Plant and machine operators and assemblers		810	(23.3)	613	(75.7)	197	(24.3)	
**Industry type**	3474							<0.001 *
Mining and quarrying		786	(22.6)	676	(86.0)	110	(14.0)	
Manufacturing		1858	(53.5)	1400	(75.3)	458	(24.7)	
Water supply, sewerage, waste management, and remediation activities		89	(2.6)	83	(93.3)	6	(6.70)	
Construction		190	(5.5)	153	(80.5)	37	(19.5)	
Wholesale and retail trade and repair of vehicles and motorcycles		9	(0.3)	8	(88.9)	1	(11.1)	
Transportation and storage		42	(1.2)	27	(64.3)	15	(35.7)	
Professional, scientific, and technical activities		84	(2.4)	80	(95.2)	4	(4.8)	
Administrative and supportive service activities		416	(12)	337	(81.0)	79	(19.0)	

ONIHL: occupational noise-induced hearing loss. * Significant at the 5% level. The categorical association is conducted using global chi-square test.

**Table 3 ijerph-18-05295-t003:** Crude and adjusted odds ratios of occupational noise-induced hearing loss among 3474 industrial workers.

Demographic Characteristics	Sub-Group	Crude Odds Ratio of ONIHL			Adjusted Odds Ratio of ONIHL		
	N	OR	(95% C.I.)	*p*-Value	AOR	(95% C.I.)	*p*-Value
**Age (years)**							
21–30	712	1	(Reference)		1	(Reference)	
31–40	1268	2.2	(1.6–3.1)	0.001 *	1.8	(1.3–2.7)	0.002 *
41–50	921	5.7	(4.0–8.1)	0.001 *	4.6	(3.2–6.7)	0.001 *
51–60	517	16.4	(11.4–23.5)	0.001 *	13.2	(8.8–19.6)	0.001 *
≥61	56	38.5	(20.1–74.0)	0.001 *	30.5	(15.2–61.5)	0.001 *
**Gender**							
Male	3434	1	(Reference)		1	(Reference)	
Female	40	0.1	(0.013–0.716)	0.022 *	0.182	(0.02–1.4)	0.100
**Nationality**							
Indian	1974	1	(Reference)		1	(Reference)	
Egyptian	483	0.1	(0.73–1.2)	0.628	1.1	(0.9–1.5)	0.346
Bangladeshi	271	1.2	(0.87–1.6)	0.301	0.9	(0.6–1.3)	0.568
Filipino	181	1.6	(1.1–2.2)	0.008 *	1.5	(1.0–2.2)	0.039 *
Pakistani	141	2.4	(1.7–3.4)	0.000 *	2.2	(1.4–3.3)	0.001 *
Kuwaiti	70	0.8	(0.44–1.6)	0.581	1.0	(0.4–2.7)	0.986
Others	354	0.7	(0.52–0.97)	0.033 *	0.8	(05–1.1)	0.178
**Work characteristics**							
Experience (years)							
≤15	2974	1	(Reference)		1	(Reference)	
16–30	418	2.5	(2.0–3.1)	0.001 *	1.1	(0.9–1.5)	0.380
≥30	49	7.9	(4.4–14.4)	0.001 *	2.2	(1.1–4.3)	0.021 *
Exposure to Noise							
No	2262	1	(Reference)		1	(Reference)	
Yes	1212	2.2	(1.9–2.6)	0.001 *	2.0	(1.7–2.4)	0.001 *
**Job and industry type**							
**Job Type**							
Elementary occupations	735	1	(Reference)		1	(Reference)	
Managers	38	2.3	(1.1–4.8)	0.024 *	1.2	(0.5–2.8)	0.673
Professionals	204	0.8	(0.5–1.3)	0.326	0.9	(0.5–1.5)	0.579
Technicians and associate professionals	886	1.2	(0.9–1.6)	0.198	0.9	(0.7–1.3)	0.641
Clerical support	110	0.7	(0.4–1.4)	0.259	0.7	(0.4–1.4)	0.350
Services and sales	22	2.6	(1.1–6.6)	0.038 *	1.5	(0.5–4.1)	0.436
Crafts and related trade works	669	2.3	(1.8–3.0)	0.000 *	1.6	(1.2–2.2)	0.002 *
Plant and machine operators and assemblers	810	1.8	(1.4–2.3)	0.000 *	0.9	(0.7–1.3)	0.737
**Industry type**							
Administrative and supportive service activities	416	1	(Reference)		1	(Reference)	
Mining and quarrying	786	0.3	(0.2–0.8)	0.024 *	0.7	(0.5–1.1)	0.116
Manufacturing	1858	0.5	(0.3–1.1)	0.013 *	0.8	(0.6–1.2)	0.347
Water supply, sewerage, waste management and remediation activities	89	0.3	(0.1–0.8)	0.008 *	0.5	(0.2–1.3)	0.161
Construction	190	1.1	(0.4–3.6)	0.888	1.3	(0.8–2.2)	0.297
Wholesale and retail trade and repair of vehicles and motorcycles	9	1.0	(0.5–2.1)	0.556	0.2	(0.01–1.5)	0.109
Transportation and storage	42	0.8	(0.4–1.6)	0.012 *	1.6	(0.8–3.8)	0.257
Professional, scientific, and technical activities	84	0.4	(0.2–0.9)	0.003 *	0.5	(0.2–1.5)	0.237

Source: SIMC Kuwait (2018). ONIHL: occupational noise-induced hearing loss; OR: odds ratio. * Significant at the 5% level. Adjusted OR = adjusted for age, gender, nationality, years of experience, noise exposure, job type, and industry type.

**Table 4 ijerph-18-05295-t004:** Summary of workplace noise regulations in Kuwait.

Regulator	Law Number	Regulation
KEPA	Environmental Protection Law 42/2014 Amended by 99/2015-Article (19)	“All establishments, in the exercise of their activities, are obligated to ensure the safety of workers and prevent exposure to damage resulting from the emission or leakage of pollutants in the work environment whether as a result of the nature of the establishment’s practice of its activities or defects in equipment. Moreover, the necessary measures include taking precautions and measures to stay within the permissible safe limits for exposure to chemicals, noise and vibration, heat and humidity, lighting and ultrasound, inactive radiation, and other requirements specified by the executive regulations of this law.”
KEPA	Environmental Protection Law 42/2014 Amended by 99/2015-Article (54)	“All parties and individuals producing or providing services, mostly during the operation of machinery and equipment and the use of alarm machines and amplifiers, are obligated to stay within the permissible limits of noise level and to conduct related activities in places allocated for this purpose. Licensing authorities should consider the use of appropriate machinery such that the total frequencies of noise emitted from fixed sources in an area stay within permissible limits.”
KEPA	Environmental Protection Law 42/2014 Amended by 99/2015-Article (55)	“The construction of establishments that emit noise and cause damage to the neighborhood environment is prohibited. The authority shall work to ensure the application of noise reduction regulations in roads, public projects, and around human gatherings and within the controls outlined in the executive regulations of this law. A fine of 500 Kuwaiti Dinars is imposed on any industrial institution found not observing this law”
KEPA	Environmental Protection Law Decision No. (210/2001)	The permissible noise levels must be less than 85 dBA per 8 h
Ministry of Social Affairs and Manpower	Ministerial Resolution No. (208/2011)	The noise level in the workplace must be less than 85 dB and should not exceed 98 dB with a maximum exposure of 8 h per day

## Data Availability

Restrictions apply to the availability of these data. Data was obtained from Shuaiba Industrial Medical Center (SIMC) and are available [from the corresponding author] with the permission of [SIMC].

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
