# Peer review of "Occupational Noise-Induced Hearing Loss among Migrant Workers in Kuwait"

_ijerph, 2021, doi:10.3390/ijerph18105295_

Round 1

Reviewer 1 Report

I have found an interesting and very complete article. I have some minors recommendations to improve the manuscript. Just review some details about the written. There are some parts using differents fonts, please check and standardize the font. Check some typos in references 20 and 23.

Reviewer 2 Report

3747 migrant workers in Kuwait measured their hearing in 2018 and occurrence of occupation induced hearing loss was related to age, work experience, subjective noise exposure, type of work and nationality. 

Many factors were related to the hearing loss, as expected. The need for worker protection is discussed.

The study is interesting and the issue is important. Many groups are of course quite small and results should be interpreted with caution. The exposure at work and at leisure time has not been separated.

The authors must clarify on the audiometric definition of the hearing loss (end page 2 to start of page 3). Three criteria, with an "or" in between. General slight 25db impairment 0.5-4khz enough? A notch, how deep? The third criterion I do not understand.

A couple of patients seem to be missing from all the tables.

Reviewer 3 Report

The manuscript from Buqammaz et al., reports the prevalence and predictors of occupational noise-induced hearing loss amongst migrant workers in Kuwait. Overall, the manuscript is well-written and presented quite clearly. Statistical reporting is excellent.  The study is limited by the fact that the conclusions are based on a single audiogram at one time-point. The authors have duly pointed out this as a limitation. I only have few minor comments:

1) The authors mentioned that audiometric tests were classified into three categories. It would help improve the manuscript, if this data is stratified and presented in a table.

2) It would also be good to see stratified data for the self-reported exposure to noise. 

3) Please provide details as to how exposure to noise was defined in the survey.
